# No association between disease severity and respiratory syncytial virus subtypes RSV-A and RSV-B in hospitalized young children in Norway

**Håkon Bøås**[1]☯*, **Lise Beier Havdal**[1,2]☯, **Ketil Størdal**[3,4], **Henrik Døllner**[5,6], **Truls Michael Leegaard**[7,8], **Terese Bekkevold**[1], **Elmira Flem**[1], **Christopher Inchley**[2], **Svein Arne Nordbø**[9,6], **Astrid Elisabeth Rojahn**[10], **Sara Debes**[11], **Bjørn Barstad**[12], **Elisebet Haarr**[13], **Anne-Marte Bakken Kran**[1,14], **for the Norwegian Enhanced Pediatric Immunisation Surveillance (NorEPIS) Network**¶

1 Norwegian Institute of Public Health, Oslo, Norway, 2 Department of Paediatric and Adolescent Medicine, Akershus University Hospital, Lørenskog, Norway, 3 Department of Pediatrics, Østfold Hospital, Grålum, Norway, 4 Division of Paediatric and Adolescent Medicine, Institute of Clinical Medicine, University of Oslo, Oslo, Norway, 5 Department of Pediatrics, St. Olavs University Hospital, Trondheim, Norway, 6 Department of Clinical and Molecular Medicine, Norwegian University of Science and Technology, Trondheim, Norway, 7 Department of Microbiology and Infection Control, Akershus University Hospital, Lørenskog, Norway, 8 Division of Medicine and Laboratory Sciences, Institute of Clinical Medicine - Campus Ahus, University of Oslo, Oslo, Norway, 9 Department of Medical Microbiology, St. Olavs University Hospital, Trondheim, Norway, 10 Division of Paediatric and Adolescent Medicine, Oslo University Hospital, Oslo, Norway, 11 Department of Medical Microbiology, Østfold Hospital, Grålum, Norway, 12 Department of Pediatrics, Stavanger University Hospital, Stavanger, Norway, 13 Department of Medical Microbiology, Stavanger University Hospital, Stavanger, Norway, 14 Department of Microbiology, Oslo University Hospital, Oslo, Norway

☯ These authors contributed equally to this work.
¶ A list of members of The Norwegian Enhanced Pediatric Immunisation Surveillance (NorEPIS) Network is provided in the Acknowledgments.
* hakon.boas@fhi.no

**Data Availability Statement:** External researchers are able to request access to data on ICD-10 codes of risk group diagnoses and respiratory diagnoses,

## Abstract

### Objective

There is conflicting evidence whether subtypes of Respiratory syncytial virus have different seasonality or are differentially associated with clinical severity. We aimed to explore the associations between disease severity and RSV subtypes RSV-A and RSV-B and to describe the circulation of RSV subtypes pattern by season and age.

### Methods

Active prospective hospital surveillance for RSV-A and RSV-B in children <59 months of age was conducted during 2015–2018. All febrile children 12–59 months of age were enrolled, whereas children <12 months were eligible if presenting with fever or respiratory symptoms. Risk factors and upper and lower respiratory tract infection was identified by linkage to national registry data and analyzed using penalized maximum likelihood logistic regression.

sex, age groups, respiratory support, hospital and admission and discharge dates from the Norwegian Patient registry. Researchers can apply for data access through Helsedata.no (https://helsedata.no/en/). Because of ethical and legal restrictions to protect the identity and confidentiality of the participants, as the data contain sensitive information according to the Norwegian Personal Data Act, all other underlying data are only available in aggregated form upon request from the Norwegian Institute of Public Health. Requests for access can be made to the data access committee represented by Anne-Marte Bakken Kran (Anne-MarteBakken.Kran@fhi.no) or by email to the Norwegian Institute of public health (folkehelseinstituttet@fhi.no, subject: Request for data access - NorEpis).

**Funding:** This work was supported by The Research Council of Norway (https://www.forskningsradet.no/en/) [240207/F20]. The funders had no role in study design, data collection and analysis, decision to publish, or preparation of the manuscript.

**Competing interests:** I have read the journal's policy and the authors of this manuscript have the following competing interests: Elmira Flem is currently employed by Merck & Co., Inc., North Wales, PA, USA. The work for the current study was conducted by Dr. Flem under the previous affiliation at the Norwegian Institute of Public Health. All other authors hereby declare that no other conflicts of interest exist.

## Results

Both RSV-A and B were found to co-circulate throughout all three study seasons, and no clear seasonal pattern was identified. Likewise, we found no association between sex or measures of severity with RSV-A or RSV-B. There was significantly more RSV-A than RSV-B among children with comorbidities.

## Conclusions

No association was found between disease severity or sex and RSV subtypes RSV-A and RSV-B in hospitalized young children in Norway.

## Introduction

Respiratory syncytial virus (RSV) is the most common cause of viral lower respiratory tract infections in infants and young children worldwide, being responsible for an estimated 33 million episodes of lower respiratory tract respiratory infections (LRTI) globally each year [1]. Children younger than one year of age, elderly or immunocompromised individuals carry a substantial burden of severe cases [2], but re-infections with mild symptoms are common in older children and healthy adults [3], indicating that infection with RSV induces partial immunity [4]. Variation in RSV strains may play a role in the ability to evade existing immunity and cause re-infections throughout life [5, 6].

RSV encompasses two major antigenic subtypes, RSV-A and RSV-B, based on the genetic variability of the G surface glycoprotein [7]. The two strains are further subdivided into several genotypes [7, 8]. Usually RSV-A and B co-circulate during the same season [8, 9], although some temporal and geographic clustering as well as alternating patterns of predominance do occur [7, 10]. Even multiple genotypes of each subtype will often co-exist during the same epidemic season [9].

There is conflicting evidence regarding the association of different RSV subtypes and clinical severity. RSV-A has been associated with more severe disease course in several studies [11–15], whereas others find subtype B to cause more severe disease [16, 17]. However, several studies report no difference in disease severity between the two subtypes [18–23].

An enhanced understanding of both epidemiology and disease severity is important for the composition of preventive strategies, as novel vaccines and monoclonal antibodies for prophylactic treatment are developed and approved. The monoclonal antibody nirsevimab for use in neonates and infants was recently approved by the U.S. Food and Drug Administration (FDA) and European Medicine Agency [24, 25], as were the first RSV vaccines for use in adults from 60 year of age and older, by the U.S. Food and Drug Administration (FDA) [26, 27]. With several maternal and pediatric vaccines against RSV currently in late-stage clinical development, defining the relationship between viral strains and clinical presentation is highly relevant and might impact decisions in the development of future protective strategies.

In a previous study, [28] we reported that 40% of Norwegian children 0–59 months of age presenting at the hospital with fever and/or respiratory symptoms during the influenza season (defined as the period between week 40 and week 20) tested positive for RSV. We found that young age, the presence of comorbidities and having siblings was associated with more severe RSV disease [29]. In the current study, we aimed to explore the associations between disease severity and RSV subtypes RSV-A and RSV-B in Norway between 2015 and 2018. We further

aimed to describe the distribution of RSV subtypes pattern by season, sex, age groups and underlying comorbidities.

## Methods

Children from 0–59 months of age presenting at hospital were prospectively enrolled through the Norwegian Enhanced Paediatric Immunisation Surveillance Network (NorEPIS). Both inpatients and outpatients were eligible. A detailed description including inclusion- and exclusion criteria has been given previously [28]. Briefly, during three seasons from 2015 to 2018, RSV surveillance was implemented annually from week 40 to week 20 the following year in five major Norwegian secondary-care hospitals, except for 2015 when surveillance was implemented from week 49. Recruitment began December 1st 2015 and ended May 21st 2018. Children between 12–59 months of age were eligible for enrollment if presenting with fever, whereas children <12 months of age were eligible if presenting with fever or respiratory symptoms. Children >5 years of age or with residence outside of the hospital's catchment area, children admitted more than 48 hours before enrollment, newborns that had not left the hospital and children referred from other hospitals or admitted for elective hospitalization, injury or social indication were excluded. A total of 2590 children 0–59 months of age were enrolled in the study and fulfilled all inclusion criteria, with a total of 2725 independent hospital contacts between them [28]. 1449 children where <12 month old and 1276 children were 12–59 months old Symptoms, clinical data, and information about healthcare use were collected using a study-specific questionnaire.

### Processing of samples

Nasopharyngeal flocked swab or aspirate samples were collected from all enrolled patients within 72 hours of arrival and analyzed at the hospital laboratory using routine in house real-time polymerase chain reactions (PCR). Three of five study hospitals used assays differentiating between RSV-A and RSV-B, in addition these hospitals tested for a respiratory panel that included Influenza, Metapneumovirus, Parainfluenza 1, 2, 3, 4 and Adenovirus. All patients with co-infections, defined as testing positive for more than one pathogen, were excluded from the analysis.

### Registry data sources

Data collected at hospital were linked to national health registries using unique personal identification numbers. The Norwegian Patient Registry (NPR) and National Health Economics Administration Database (KUHR) together cover all governmental-funded health care in Norway. NPR contains information on all hospital visits in Norway, including International Classification of Diseases (ICD-10) diagnoses [30]. KUHR contains International Classification of Primary Care (ICPC-2) or ICD-10 diagnoses from all publicly funded GPs and primary care emergency clinics [31]. The Medical Birth Registry of Norway (MBRN) contains information on gestational age, congenital malformations and disorders for all children born in Norway [32]. A detailed description of all information retrieved from the above registries for identifying underlying risk groups has been published previously [28, 33].

### Risk groups and severity

We identified children in high-risk groups using the data from MBRN, ICD-10 codes from the NPR and ICPC-2 codes from KUHR. Children were categorized as high-risk group if they were registered with an ICD-10 code/ICPC-2 code corresponding to premature birth, trisomy

21, congenital heart disease, pulmonary disease, (S1 Table and [29]). Prematurity as risk factor was defined as gestational age (GA) < 37 weeks. Two primary outcome measures were used to compare disease severity: 1) hospital length-of-stay (LOS) measured in hours from arrival, and 2) the need for respiratory support, in the form of either invasive ventilation, Continuous Positive Airway Pressure (CPAP), or Bi- level Positive Airway Pressure (BiPAP). Acute upper respiratory tract infection (URTI) or lower respiratory tract infection (LRTI) was categorized based on ICD-10 codes (S2 Table).

## Statistical analyses

Statistical analysis was performed in Stata version 15 (StataCorp LLC, College Station, Texas, US). To account for groups with low number of events, we compared the percentage of RSV-A and B positive samples using penalized maximum likelihood logistic regression to calculate odds ratios (OR) with a 95% confidence level. For comparison we also calculated OR and 95% confidence intervals using a standard logistic regression (S3 and S4 Tables). To account for potential differences in age distribution and recruitment procedure, the age groups, month of hospital contact, and the treating hospital were included as independent variables in adjusted analyzes.

The analysis involved only patients where the first reported symptoms arose in the 10 days up to hospitalization. If a patient was recruited on more than one hospital visit within a period of 21 days, and the physician assessed the cause to be the same disease episode, only the first hospital encounter was used. We compared LOS between children with RSV-A and B subtypes, using Wilcoxon ranksum test.

## Ethics

Written informed consent was obtained from both parents/legal guardians for all children. The study was reviewed and approved by the Regional committees for medical and health research ethics—South-East A (2015/956).

## Results

### Study population

During three seasons of active hospital surveillance, a total of 1096 children 0–59 months of age tested positive for RSV, of these, 815 (74%) samples were further typed and characterized as RSV-A or RSV-B. In total, 370 (45.4%) samples were positive for RSV-A and 426 (52.3%) for RSV-B. Nineteen patients (2.3%) were co-infected with both subtypes (Table 1) and were excluded from further analyses. Another 23 samples were excluded due to inconclusive or missing test results for one or more of the viruses tested for as a part of the hospitals'

**Table 1. Distribution of RSV subtypes by participant age.**

|  | 0-3m | 3-6m | 6-12m | 1-2y | 2-5y |
|---|---|---|---|---|---|
| **Negative n (%)** | 331(52.4) | 167 (48.0) | 299 (63.8) | 473 (61.4) | 359 (70.9) |
| RSV-A n (%) | 89 (14.1) | 62 (17.8) | 69 (14.7) | 100 (13.0) | 50 (9.9) |
| RSV-B n (%) | 116(18.4) | 70 (20.1) | 65 (13.9) | 113 (14.7) | 62 (12.3) |
| RSV-A+B n (%) | 4(0.6) | 2 (0.6) | 0 (0.00) | 8 (1.0) | 5 (1.0) |
| RSV not typed n (%) | 92(14.6) | 47 (13.5) | 36 (7.7) | 76 (9.9) | 30 (5.9) |
| Total RSV n (%) | 301 (47.6) | 181 (52.0) | 170 (36.2) | 297 (38.6) | 147 (29.1) |
| **Total (n = 2725.00)** | 632 (100) | 348 (100) | 469 (100) | 770 (100) | 506 (100) |

respiratory panels. Among the remaining 773 RSV positive samples, 119 (15.4%) had co-infections with other respiratory viruses and were excluded from the analysis, resulting in a total of 654 RSV positive samples analyzed (Fig 1). The study included 379 boys and 275 girls <5 years of age. We found no significant differences between the proportion of RSV-A or B by age or

NorEPIS RSV-A and RSV-B flowchart

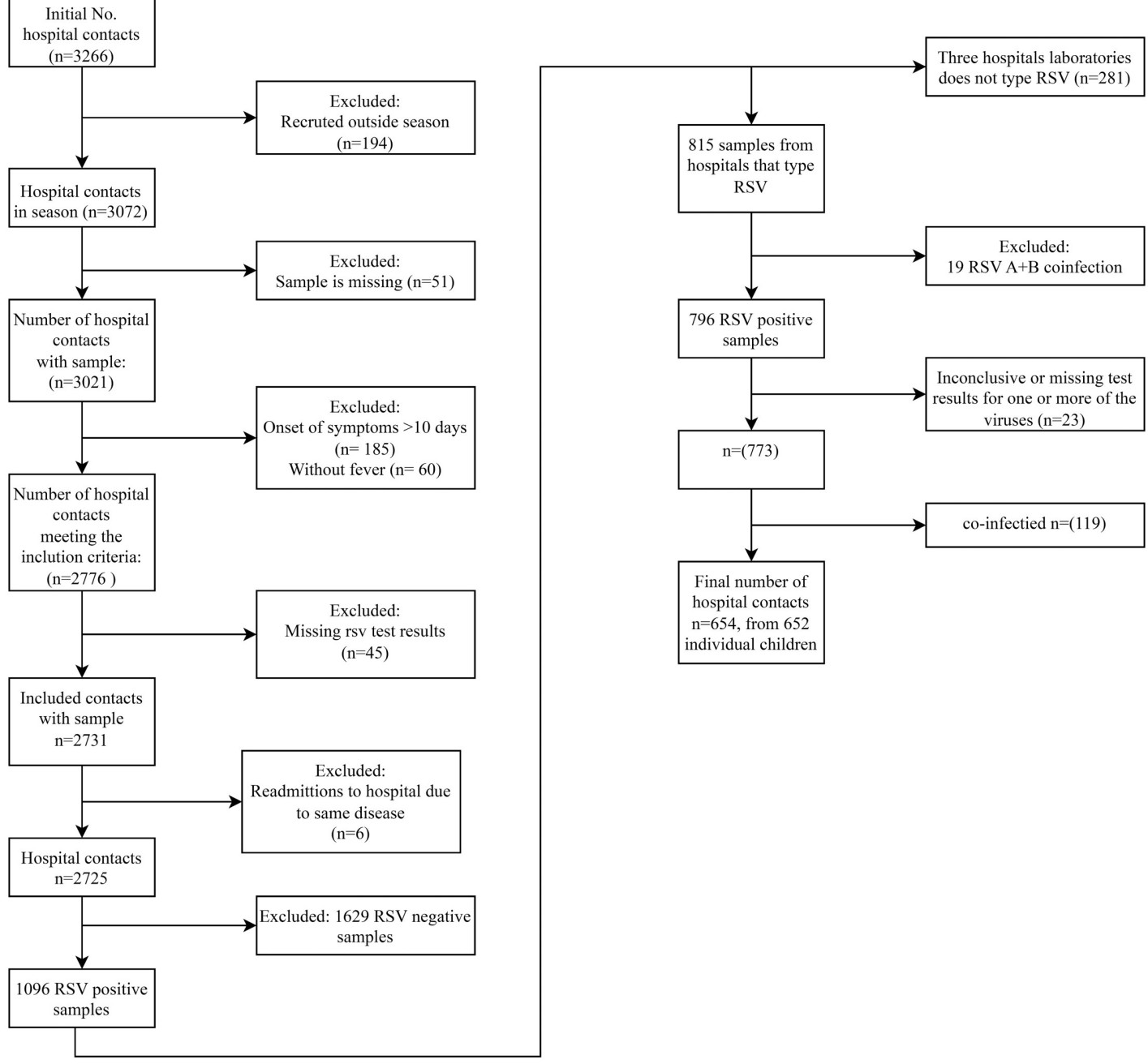

**Fig 1. Flowchart of recruitment and testing.**

sex, neither for the entire study population (Table 2) nor when restricting the analysis to inpatients only (Table 3).

## Seasonality

The number of identified RSV cases varied considerably between seasons, with the most cases identified during the 2016/17-season. Both RSV-A and B co-circulated all three seasons, with a slightly higher proportion of RSV-B in the 2015/16 season. In contrast, there was little or no difference in the occurrence of the two subtypes during the 2016/17 and 2017/18 seasons (Fig 2 & Table 2).

## Severity

We found no difference between the RSV subtypes and being admitted as an inpatient or outpatient (p = 0.99, Table 2), nor between patients admitted for more than 24 hours compared to patients with a shorter hospital stay (p = 0.75, Table 2). In the analysis of inpatients only, we found no significant difference in LOS between RSV subtypes. The median LOS for children with RSV-A were 62h (interquartile range (IQR) 27-110h) compared to median LOS 63h (IQR 32–102) for children with RSV-B (z = -0.1, p = 0.91). (Fig 3 and Table 3). There were 6 patients with a LOS exceeding 12 days. To ensure that potential errors in the registration of the admission date or discharge date did not influence the results a secondary analysis was conducted. Excluding these outliers did not affect the result (not shown). Similarly, we found no difference in need for respiratory support (p = 0.99 for all patients [Table 2] and p = 0.70 for inpatients [Table 3]) between the RSV subtypes. Furthermore, the occurrence of RSV-A and RSV-B in children diagnosed with acute upper respiratory tract infection (URTI) or lower respiratory tract infection (LRTI) was similar (p = 0.79 and p = 0.91 respectively for all patients [Table 2] and p = 0.63 and p = 0.63 respectively for inpatients [Table 3]).

## Risk factors and comorbidities

The number of children in each group of separate co-morbidities was too small for a meaningful analysis. However, there was more RSV-A (~60%) compared to RSV-B (~40%) among children with any comorbidities (Including trisomy 21, neuromuscular impairment, congenital heart disease, pulmonary disease, BPD, immunodeficiency, and cancer), both among all hospital contacts (Table 2) and inpatients (Table 3). There was no association observed between prematurity and RSV subtypes or between having congenital heart disease or pulmonary disease and types of RSV (Tables 2 and 3).

## Discussion

In the current study we investigated differences in disease severity, risk factors and comorbidities between RSV-A and RSV-B subtypes in young, hospitalized children in Norway during pre-pandemic period. Both RSV-A and B was found to co-circulate throughout all three study seasons, and no clear difference in the seasonal pattern between the RSV subtypes was identified. Likewise, we found no association between age, sex or measures of severity with RSV-A or RSV-B. There was more RSV-A than RSV-B among children with comorbidities, however with a limited number of children in the study the results should be explored in further studies and currently need to be interpreted with caution.

The seasonal patterns of the RSV subtypes in circulation in Norway, has to our knowledge, not previously been described. However, the overall total RSV seasonality is similar to previous reports from Norway [34] and is presented in detail in our previous publication [28]. The co-

**Table 2. Participant (inpatients and outpatients combined) characteristics and penalized maximum likelihood logistic regression of typed RSV-cases without co-infection\*.**

| | n RSV-A/B (%B) | OR (95% Ci) | p | Adjusted OR (95% Ci) [†] | Adjusted p-value[†] |
|---|---|---|---|---|---|
| **Age group** | | | | | |
| 0-3m | 75/109 (59.2) | Ref. | | Ref. | |
| 3-6m | 53/64 (54.7) | 0.83 (0.52–1.32) | 0.437 | 0.86 (0.54–1.39) | 0.548 |
| 6-12m | 55/49 (47.1) | 0.61 (0.38–1.00) | 0.048 | 0.62 (0.38–1.01) | 0.053 |
| 1-2y | 66/94 (58.8) | 0.98 (0.64–1.51) | 0.926 | 1.06 (0.68–1.65) | 0.788 |
| 2-5y | 42/47 (52.8) | 0.77 (0.46–1.28) | 0.314 | 0.87 (0.52–1.46) | 0.591 |
| **Sex** | | | | | |
| Male | 170/209 (55.1) | Ref. | | Ref. | |
| Female | 121/154 (56.0) | 1.03 (0.76–1.41) | 0.829 | 1.01 (0.73–1.39) | 0.974 |
| **Hospital** | | | | | |
| Ullevål | 153/159 (51.0) | Ref. | | Ref. | |
| AHUS | 62/85 (57.8) | 1.32 (0.89–1.95) | 0.172 | 1.34 (0.90–2.01) | 0.152 |
| Østfold | 76/119 (61.0) | 1.50 (1.05–2.16) | 0.028 | 1.49 (1.03–2.16) | 0.036 |
| **Study season** | | | | | |
| 2015/2016 | 80/144 (64.3) | Ref. | | Ref. | |
| 2016/2017 | 165/159 (49.1) | 0.54 (0.38–0.76) | <0.001 | 0.57 (0.38–0.87) | 0.009 |
| 2018/2019 | 46/60 (56.6) | 0.72 (0.45–1.16) | 0.179 | 0.80 (0.49–1.31) | 0.377 |
| **Patient type** | | | | | |
| Out-patient | 127/152 (54.5) | Ref. | | Ref. | |
| Inpatient | 160/205 (56.2) | 1.07 (0.78–1.46) | 0.670 | 1.00 (0.72–1.39) | 0.985 |
| **Length of stay** | | | | | |
| <24 hours | 162/192 (54.2) | Ref. | | Ref. | |
| > = 24 hours | 125/165 (56.9) | 1.11 (0.81–1.52) | 0.501 | 1.05 (0.76–1.46) | 0.754 |
| **Respiratory support[‡]** | | | | | |
| No | 243/305 (55.7) | Ref. | | Ref. | |
| Yes | 42/53 (55.8) | 1.00 (0.65–1.55) | 0.988 | 1.00 (0.63–1.59) | 0.990 |
| **Acute upper respiratory tract infection (URTI)** | | | | | |
| No | 238/292 (55.1) | Ref. | | Ref. | |
| Yes | 53/71 (57.3) | 1.09 (0.74–1.61) | 0.669 | 1.06 (0.70–1.59) | 0.794 |
| **Lower respiratory tract infection (LRTI)** | | | | | |
| No | 68/90 (57.0) | Ref. | | Ref. | |
| Yes | 223/273 (55.0) | 0.93 (0.65–1.33) | 0.677 | 0.98 (0.67–1.43) | 0.911 |
| **Congenital heart disease or pulmonary disease (including BPD)** | | | | | |
| No | 268/344 (56.2) | Ref. | | Ref. | |
| Yes | 23/19 (45.2) | 0.65 (0.35–1.20) | 0.169 | 0.64 (0.34–1.23) | 0.180 |
| **Comorbidities[§]** | | | | | |
| No | 255/340 (57.1) | Ref. | | Ref. | |
| Yes | 36/23 (39.0) | 0.48 (0.28–0.83) | 0.009 | 0.47 (0.27–0.83) | 0.009 |
| **Gestational age <37 weeks** | | | | | |
| No | 255/322 (55.8) | Ref. | | Ref. | |
| Yes | 36/41 (53.3) | 0.90 (0.56–1.45) | 0.666 | 0.88 (0.54–1.44) | 0.619 |

\* Patients co-infected with one or more of Influenza, Metapneumovirus, Parainfluenza 1, 2, 3, 4 and Adenovirus, or cases with missing information about coinfections are excluded from this table.

[†] Age groups, month of hospital contact, and the treating hospital were included as independent variables

[‡] Invasive ventilation, Continuous Positive Airway Pressure (CPAP), or Bi-level Positive Airway Pressure (BiPAP)

[§] Including trisomy 21, neuromuscular, impairment, congenital heart disease, pulmonary disease, BPD, immunodeficiency, and cancer.

**Table 3. Inpatients characteristics and penalized maximum likelihood logistic regression of typed RSV-cases without co-infection*.**

| | n RSV-A/B (%B) | OR (95% Ci) | p | Adjusted OR (95% Ci) [†] | Adjusted p-value[†] |
|---|---|---|---|---|---|
| **Age group** | | | | | |
| 0-3m | 53/79 (59.9) | Ref. | | Ref. | |
| 3-6m | 23/41 (64.1) | 1.19 (0.64–2.19) | 0.581 | 1.31 (0.70–2.46) | 0.397 |
| 6-12m | 24/18 (42.9) | 0.51 (0.25–1.02) | 0.057 | 0.55 (0.27–1.11) | 0.096 |
| 1-2y | 36/45 (55.6) | 0.84 (0.48–1.46) | 0.536 | 1.00 (0.56–1.78) | 0.993 |
| 2-5y | 24/22 (47.8) | 0.62 (0.32–1.21) | 0.159 | 0.73 (0.37–1.45) | 0.373 |
| **Sex** | | | | | |
| Male | 84/118 (58.4) | Ref. | | Ref. | |
| female | 76/87 (53.4) | 0.82 (0.54–1.23) | 0.335 | 0.78 (0.51–1.20) | 0.264 |
| **Hospital** | | | | | |
| Ullevål | 85/98 (53.5) | Ref. | | Ref. | |
| AHUS | 35/47 (57.3) | 1.16 (0.69–1.96) | 0.575 | 1.20 (0.70–2.04) | 0.510 |
| Østfold | 40/60 (60.0) | 1.30 (0.79–2.12) | 0.301 | 1.28 (0.77–2.13) | 0.337 |
| **Study season** | | | | | |
| 2015/2016 | 44/77 (63.6) | Ref. | | Ref. | |
| 2016/2017 | 93/91 (49.5) | 0.56 (0.35–0.90) | 0.016 | 0.55 (0.32–0.95) | 0.032 |
| 2018/2019 | 23/37 (61.7) | 0.92 (0.49–1.73) | 0.787 | 0.95 (0.49–1.83) | 0.873 |
| **Length of stay** | | | | | |
| <24 hours | 35/40 (53.3) | Ref. | | Ref. | |
| > = 24 hours | 125/165 (56.9) | 1.16 (0.70–1.92) | 0.575 | 1.17 (0.70–1.96) | 0.543 |
| **Respiratory support[‡]** | | | | | |
| No | 121/152 (55.7) | Ref. | | Ref. | |
| Yes | 36/50 (58.1) | 1.10 (0.68–1.80) | 0.696 | 1.11 (0.66–1.86) | 0.700 |
| **Acute upper respiratory tract infection (URTI)** | | | | | |
| No | 146/189 (56.4) | Ref. | | Ref. | |
| Yes | 14/16 (53.3) | 0.88 (0.42–1.84) | 0.734 | 0.83 (0.39–1.76) | 0.629 |
| **Lower respiratory tract infection (LRTI)** | | | | | |
| No | 15/16 (51.6) | Ref. | | Ref. | |
| Yes | 145/189 (56.6) | 1.22 (0.59–2.53) | 0.586 | 1.20 (0.56–2.56) | 0.633 |
| **Congenital heart disease or pulmonary disease (including BPD)** | | | | | |
| No | 145/193 (57.1) | Ref. | | Ref. | |
| Yes | 15/12 (44.4) | 0.61 (0.28–1.32) | 0.206 | 0.62 (0.28–1.38) | 0.244 |
| **Comorbidity[§]** | | | | | |
| No | 136/190 (58.3) | Ref. | | Ref. | |
| Yes | 24/15 (38.5) | 0.45 (0.23–0.89) | 0.021 | 0.49 (0.25–0.99) | 0.046 |
| **Gestational age <37 weeks** | | | | | |
| No | 140/180 (56.3) | Ref. | | Ref. | |
| Yes | 20/25 (55.6) | 0.97 (0.52–1.80) | 0.919 | 1.04 (0.54–1.98) | 0.917 |

* Patients co-infected with one or more of Influenza, Metapneumovirus, Parainfluenza 1, 2, 3, 4 and Adenovirus, or cases with missing information about coinfections are excluded from this table.

[†] Age groups, month of hospital contact, and the treating hospital were included as independent variables

[‡] Invasive ventilation, Continuous Positive Airway Pressure (CPAP), or Bi-level Positive Airway Pressure (BiPAP)

[§] Including trisomy 21, neuromuscular, impairment, congenital heart disease, pulmonary disease, BPD, immunodeficiency, and cancer.

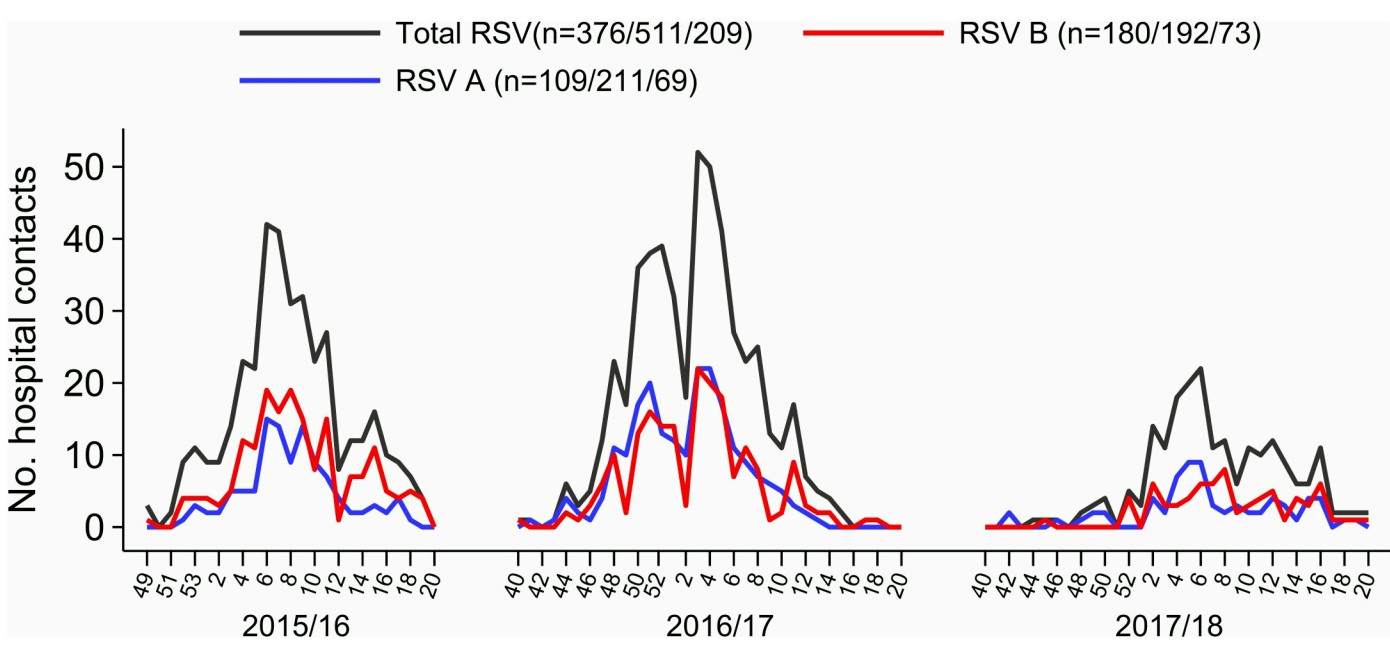

**Fig 2. Number of detected and subtyped RSV positive samples per week through three seasons.**

circulation of RSV-A and RSV-B was expected and in line with previous reports [8, 9]. Although RSV-B was slightly more dominant during the 2015/16 season, there was no clear dominant subtype during the following two seasons, and little evidence was found for alternating patterns or a strong predomination of any subtype, as has been found by some [7, 10]. During the COVID-19 pandemic, many countries reported a disrupted seasonality of RSV, with little or no RSV circulating during the first years of the pandemic and with untypical out of seasonal peaks once disease control measures was lifted [35]. This disrupted and shifted seasonality of RSV was also observed in Norway during and after the pandemic [36]. Future studies are needed to elucidate if there are differences in the distribution of RSV subtypes after the pandemic. The duration of this study, spanning three seasons, is not ideal for studying shifting seasonal pattern. Thus the lack of predominance of one subtype or different seasonal patterns of RSV subtypes in this study of hospitalized cases, does not exclude the possibility of such a pattern in the community. Although efforts were made to exclude coinfections with common pathogenic viruses, we did not test for all known respiratory viruses, and it is possible that coinfections with respiratory agents such as bocavirus, or endemic coronaviruses could have a limited impact on the results.

### Age and sex

We found no difference in sex and age distribution between the two RSV subtypes, and there does not seem to be any predominance of any of the RSV subtypes between age groups [37–39]. One study by Tabor, et al. [7] reported the global proportion of RSV-B to be higher in females than males; however, this lacks support in most other studies [7, 13, 15, 38, 39], which is consistent with our results [7, 13, 15, 38, 39].

### Severity

Several studies have investigated differences in disease severity of RSV-A and B, but the results remain inconclusive (reviewed in [6]). One study from Italy found RSV-A to predominate in

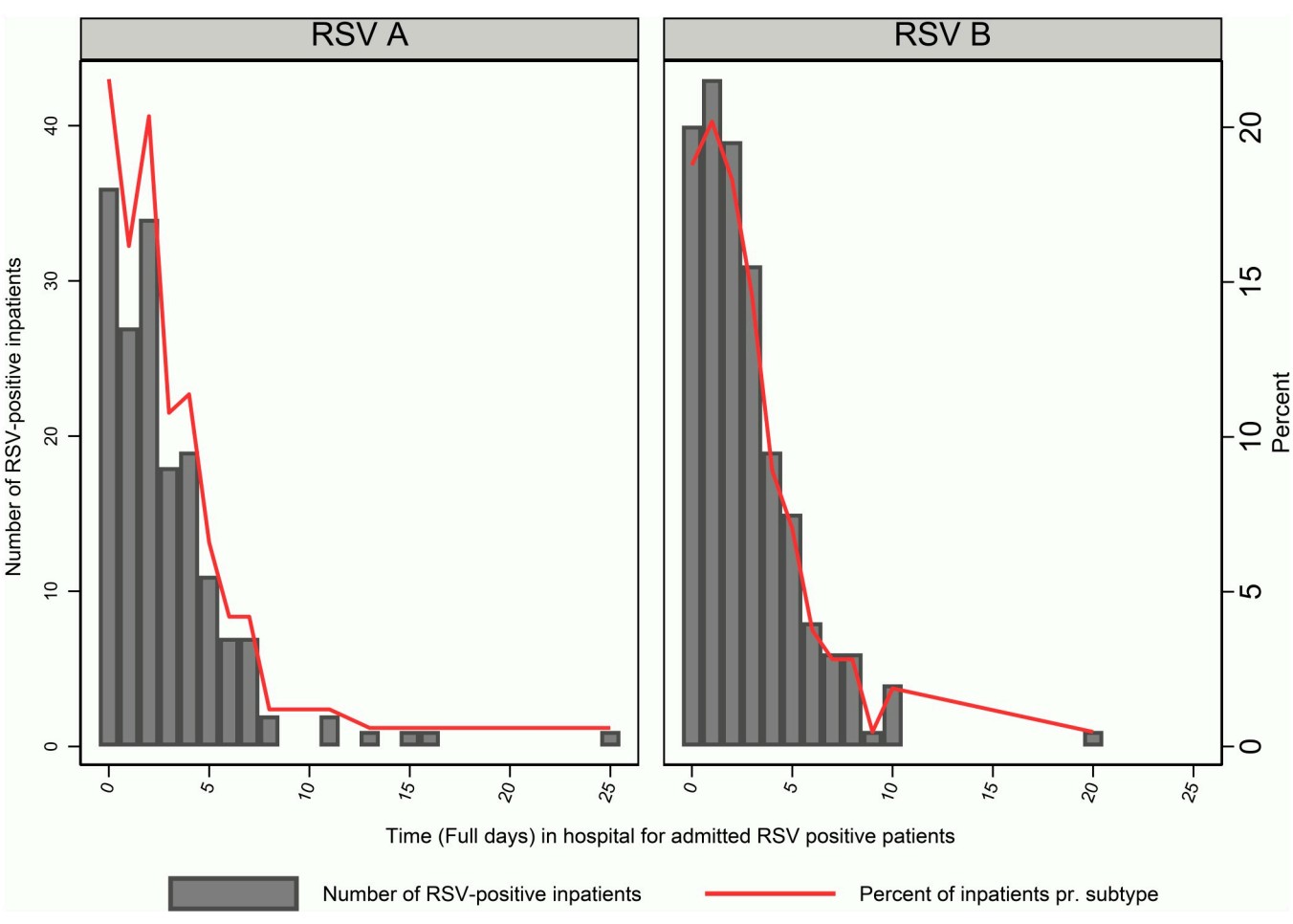

**Fig 3. Duration of hospital stay by RSV subtype positivity (number and percentage).**

severe cases among premature children [21], whereas community-based monitoring from rural India [40] found RSV-B to cause more severe lower respiratory tract infections in pre-term babies. In our study, we found no association between the RSV subtypes and proxies for severity such as length of stay among inpatients, treatment with either of CPAP, BiPAP or high-flow oxygen treatment, or between children diagnosed with URTI or LRTI (Tables 2 and 3). It is not uncommon for children presenting at hospital with RSV to be sent home on the first assessment, and later return to hospital if the condition worsens. Some patients are admitted several times during the course of an RSV infection. In order to avoid counting these patients multiple times we only used the first registered hospital contact during a disease episode, potentially failing to capture the full duration of combined hospital stays, and subsequent loss of power to detect associations between RSV subtypes and length of stay or respiratory treatment. However, about 88% of all RSV cases in the NorEpis study had only one hospital contact during the disease episode [41]. There does not seem to be a predominance of either of the RSV subtypes among premature children with GA < 37 weeks. Restricting the analysis of premature children to extremely premature or very premature children according to the WHO definitions [42], could be warranted, however this was not possible in this study due to low numbers of premature children among the RSV-A and RSV-B cases.

### Risk factors and comorbidities

A presence of comorbidities (trisomy 21, neuromuscular impairment, congenital heart disease, pulmonary disease, BPD, immunodeficiency, or cancer) in our previous study was associated with the need for respiratory support and a duration of stay of >72h when hospitalized with RSV [29]. Here we demonstrate that there was significantly more RSV-A than RSV-B among children with comorbidities. To our knowledge, this is the first study to investigate if there is an association between the presence of comorbidities and RSV subtypes. However, one study found a stronger association between RSV-B infection and a family history of asthma compared to RSV-A. However, this association also varied between RSV-A genotypes, with some genotypes showing a similar association to asthma as RSV-B [43]. The number of RSV positive children with comorbidities was generally very low, making the study underpowered to detect differences between the RSV subtypes and individual comorbidities. The low number of events per risk group makes the estimates vulnerable to random patterns. Therefore, caution should be taken when interpreting the distribution of RSV subtypes in our study. It is possible that any association between risk factors and RSV subtypes are more dependent on different genotypes, than on the subtypes themselves [6, 43], which should be considered in future studies.

### Conclusion

No association was found between disease severity or sex and RSV subtypes RSV-A and RSV-B in hospitalized young children in Norway. However, the higher proportion of RSV-A in children with co-morbidities warrants further studies. Both RSV subtypes co-circulate and there are no evident differences in severity. Future studies are needed to investigate how the distribution of RSV and RSV subtypes has changed during and after the COVID-19 pandemic, and the potential impact this has for the prevention of RSV in the future.

### Supporting information

**S1 Table. ICD-10 and ICPC-2 codes used to identify children with high risk of severe Influenza.** When no additional cases were identified by ICPC-2 codes, only ICD-10 codes are presented.
(DOCX)

**S2 Table. ICD-10 codes used to identify relevant diagnosis groups.**
(DOCX)

**S3 Table. Characteristics and logistic regression of typed RSV-cases.** Excluding co-infections with one or more of Influenza, Metapneumovirus, Parainfluenza 1, 2, 3, 4 and Adenovirus, or cases with missing information on coinfections.
(DOCX)

**S4 Table. Inpatient characteristics and logistic regression of typed RSV-cases.** Excluding co-infections with one or more of Influenza, Metapneumovirus, Parainfluenza 1, 2, 3, 4 and Adenovirus, or cases with missing information on coinfections.
(DOCX)

### Acknowledgments

We would like to express our sincere gratitude to the participants, and the personnel involved in the NorEPIS network and respiratory project at the Norwegian Institute of Public Health. Especially all personnel at the virology department for their help in receiving and registration

samples, and the microbiology departments in the study hospitals for adapting their routines to the study. We would also like to express our gratitude to all the study nurses and hospital personnel. They have done an invaluable effort for recruitment and conducting patient interviews. The following researchers are a part of NorEPIS network:

Håkon Bøås[1]*, Lise Beier Havdal[1,2], Ketil Størdal[5,12], Henrik Døllner[7,9], Truls Michael Leegaard[13,14], Terese Bekkevold[1], Elmira Flem[1], Christopher Inchley[2], Svein Arne Nordbø[8,9], Astrid Elisabeth Rojahn[4], Sara Debes[6], Bjørn Barstad MD[10], Elisebet Haarr[11], Liliana Vázquez Fernández[1], Olav Hungnes[1], Britt Nakstad[2/4], Anne-Marte Bakken Kran[1,3],

1. Norwegian Institute of Public Health, Oslo, Norway

2. Department of Paediatric and Adolescent Medicine, Akershus University Hospital, Lørenskog, Norway

3. Department of Microbiology, Oslo University Hospital, Ullevål, Oslo, Norway

4. Division of Paediatric and Adolescent Medicine, Oslo University Hospital, Ullevål, Oslo, Norway

5. Department of Pediatrics, Østfold Hospital, Kalnes, Grålum, Norway

6. Department of Medical Microbiology, Østfold Hospital, Kalnes, Grålum, Norway

7. Department of Pediatrics, St. Olavs University Hospital, Trondheim, Norway

8. Department of Medical Microbiology, St. Olavs University Hospital, Trondheim, Norway

9. Department of Clinical and Molecular Medicine, Norwegian University of Science and Technology

10. Department of Pediatrics, Stavanger University Hospital, Stavanger, Norway

11. Department of Medical Microbiology, Stavanger University Hospital, Stavanger, Norway

12. Division of Paediatric and Adolescent Medicine, Institute of Clinical Medicine, University of Oslo, Oslo, Norway

13. Department of Microbiology and Infection Control, Akershus University Hospital, Lørenskog

14. Institute of Clinical Medicine—Campus Ahus, Division of Medicine and Laboratory Sciences, University of Oslo, Oslo

* E-mail address: hakon.boas@fhi.no (H. Bøås)

## Author Contributions

**Conceptualization:** Håkon Bøås, Lise Beier Havdal, Ketil Størdal, Henrik Døllner, Truls Michael Leegaard, Terese Bekkevold, Elmira Flem, Christopher Inchley, Svein Arne Nordbø, Astrid Elisabeth Rojahn, Sara Debes, Bjørn Barstad, Elisebet Haarr, Anne-Marte Bakken Kran.

**Data curation:** Håkon Bøås, Lise Beier Havdal, Terese Bekkevold.

**Formal analysis:** Håkon Bøås.

**Funding acquisition:** Elmira Flem.

**Investigation:** Håkon Bøås, Elmira Flem.

**Methodology:** Håkon Bøås, Elmira Flem.

**Project administration:** Håkon Bøås, Ketil Størdal, Henrik Døllner, Truls Michael Leegaard, Terese Bekkevold, Elmira Flem, Christopher Inchley, Svein Arne Nordbø, Astrid Elisabeth Rojahn, Sara Debes, Bjørn Barstad, Elisebet Haarr, Anne-Marte Bakken Kran.

**Resources:** Terese Bekkevold, Elmira Flem, Anne-Marte Bakken Kran.

**Supervision:** Ketil Størdal, Henrik Døllner, Truls Michael Leegaard, Terese Bekkevold, Elmira Flem, Christopher Inchley, Svein Arne Nordbø, Astrid Elisabeth Rojahn, Sara Debes, Bjørn Barstad, Elisebet Haarr.

**Validation:** Håkon Bøås.

**Visualization:** Håkon Bøås, Anne-Marte Bakken Kran.

**Writing – original draft:** Håkon Bøås, Lise Beier Havdal.

**Writing – review & editing:** Håkon Bøås, Lise Beier Havdal, Ketil Størdal, Henrik Døllner, Truls Michael Leegaard, Terese Bekkevold, Elmira Flem, Christopher Inchley, Svein Arne Nordbø, Astrid Elisabeth Rojahn, Sara Debes, Bjørn Barstad, Elisebet Haarr, Anne-Marte Bakken Kran.

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
