## [Decision Letter · Decision Letter 0]

9 Nov 2023

PONE-D-23-21221No association between disease severity and Respiratory syncytial virus subtypes RSV-A and RSV-B in hospitalized young children in NorwayPLOS ONE

Dear Dr. Bøås,

Thank you for submitting your manuscript to PLOS ONE. After careful consideration, we feel that it has merit but does not fully meet PLOS ONE’s publication criteria as it currently stands. Therefore, we invite you to submit a revised version of the manuscript that addresses the points raised during the review process.

We look forward to receiving your revised manuscript.

Kind regards,

Victor Daniel Miron

Academic Editor

PLOS ONE

Journal Requirements: 

"I have read the journal's policy and the authors of this manuscript have the following competing interests: Elmira Flem is currently employed by Merck & Co., Inc., North Wales, PA, USA. The work for the current study was conducted by Dr. Flem under the previous affiliation at the Norwegian Institute of Public Health. 

All other authors hereby declare that no other conflicts of interest exist."

We note that one or more of the authors are employed by a commercial company: Merck & Co., Inc. 

(2) Please also provide an updated Competing Interests Statement declaring this commercial affiliation along with any other relevant declarations relating to employment, consultancy, patents, products in development, or marketed products, etc.  

Within your Competing Interests Statement, please confirm that this commercial affiliation does not alter your adherence to all PLOS ONE policies on sharing data and materials by including the following statement: "This does not alter our adherence to  PLOS ONE policies on sharing data and materials.” (as detailed online in our guide for authors http://journals.plos.org/plosone/s/competing-interests).

If this adherence statement is not accurate and  there are restrictions on sharing of data and/or materials, please state these. Please note that we cannot proceed with consideration of your article until this information has been declared.

4. One of the noted authors is a group or consortium: Norwegian Enhanced Pediatric Immunisation Surveillance (NorEPIS) Network

In addition to naming the author group, please list the individual authors and affiliations within this group in the acknowledgments section of your manuscript. Please also indicate clearly a lead author for this group along with a contact email address.

Reviewers' comments:

Reviewer's Responses to Questions

**Comments to the Author**

1. Is the manuscript technically sound, and do the data support the conclusions?

Reviewer #1: Yes

Reviewer #2: Yes

2. Has the statistical analysis been performed appropriately and rigorously? 

Reviewer #1: Yes

Reviewer #2: I Don't Know

3. Have the authors made all data underlying the findings in their manuscript fully available?

Reviewer #1: Yes

Reviewer #2: Yes

4. Is the manuscript presented in an intelligible fashion and written in standard English?

Reviewer #1: Yes

Reviewer #2: No

5. Review Comments to the Author

Reviewer #1: Dear colleagues I would like to thank you for the interesting research you have performed.

I was impressed by the high volume of data collected and the meticulous analysis performed.

Only trivial comments and questions:

1. page 11 first row of "Table 2: Participant characteristics and penalized maximum likelihood logistic regression of typed RSV cases. Excluding co-infections with one or more of Influenza, Metapneumovirus, Parainfluenza 1, 2, 3, 4 and Adenovirus, or cases with missing information on coinfections" - you have to correct "Inn-patients" with "inpatients"

2. row 222-3 - " Excluding 6 outliers with a LOS exceeding 12 days did not impact the result (not shown)." Please elaborate on these outliers with prolonged LOS - were they with nosocomial infections or preexisting severe conditions, etc?

3. in page 17 row 241-244 "In the current study we investigated differences in disease severity, risk factors and comorbidities between RSV-A and RSV-B subtypes in young, hospitalized children in Norway during pre-pandemic period. Both RSV-A and B was found to co-circulate throughout all three study seasons, and no clear seasonal pattern was identified."

Please elaborate on present research directions... Do you have any research ongoing on COVID-19 pandemic impact on RSV circulation and RSV subtypes? Because there is a plethora of published papers that are documenting major impact of SARS-CoV-2 phenomenon on circulation of certain viruses like influenza, RSV, etc and virtually no impact on Rhinovirus circulation. It would add value on Discussion section.

4. page 18 row 280 " we only had one hospital contact pr. disease episode" - please rephrase this statement, or better the entire section because "contact pr. disease" is not included in your abbreviations section nor is a usual abbreviation.

5. page 18 rows 285-288. Please elaborate on Norwegian national guidelines prophylaxis with palivizumab in premature children because the reference inserted [39] is not in English and can generate confusion. In your country there are implemented guidance for RSV prophylaxis in premature children of what gestational age and since when? Because study period is 2015-18 and reference provided seems to be from 2021.

Reviewer #2: Abstract:

Authors reported the results of sex association with RSV-A or RSV-B, and the comorbidities prevalence among the two types. However, these results weren’t mentioned as an objective or an aim for this study. Results of age association with RSV-A or RSV-B was meant to be an objective of the study, but no result was reported in the results section.

Introduction: Introduction is well written. However, adding more details about the conducted study is important such as the duration and location of the study.

Methods:

- Exclusion criteria wasn’t clear, try to add a paragraph indicating the exclusion criteria of your study.

- Did the studied patients were encountered at the emergency room or a clinical setting? Were they encountered in a tertiary hospital, secondary hospital, or primary care services?

- Among the enrolled 2590 children, how many of them are below the age 12 months? And how many were between 12-59 months of age?

- More details in the method section are needed, exclusion criteria wasn’t clear, please review the comments above.

- Results:

- Results (Table 2, 3): Tables are confusing, and legends does not differentiate them. Also, try to use some in text explanation of these two tables. Furthermore, what type of respiratory support? What are the comorbidities?

- In the severity section add the p-value for significance.

- More in-text elaboration of the results is needed. However, the first part of the result and the tables are confusing and weren’t clearly delivered.

Discussion

- A summarization of the main results with its significant level wasn’t included.

- A comparison of the virus seasonality with the previous findings of the virus pattern in Norway needs to be done.

- The study results of Risk factors and comorbidities weren’t compared with other studies.

- Overall and apart from the severity section, the discussion is unorganized, poorly written, and lacks the inclusion of supporting evidence and other studies.

6. PLOS authors have the option to publish the peer review history of their article (what does this mean?). If published, this will include your full peer review and any attached files.

Reviewer #1: **Yes: **Mihai Craiu

Reviewer #2: **Yes: **Dr. Mohammad Faraj Alqahtani

---

## [Author Response · Author response to Decision Letter 0]

23 Dec 2023

Journal Requirements: 

1. Please ensure that your manuscript meets PLOS ONE’s style requirements, including those for file naming. The PLOS ONE style templates can be found at 

We have changed the manuscript in accordance with the guidelines.

“I have read the journal’s policy and the authors of this manuscript have the following competing interests: Elmira Flem is currently employed by Merck & Co., Inc., North Wales, PA, USA. The work for the current study was conducted by Dr. Flem under the previous affiliation at the Norwegian Institute of Public Health. 

All other authors hereby declare that no other conflicts of interest exist.”

We note that one or more of the authors are employed by a commercial company: Merck & Co., Inc. 

(1) Please provide an amended Funding Statement declaring this commercial affiliation, as well as a statement regarding the Role of Funders in your study. If the funding organization did not play a role in the study design, data collection and analysis, decision to publish, or preparation of the manuscript and only provided financial support in the form of authors’ salaries and/or research materials, please review your statements relating to the author contributions, and ensure you have specifically and accurately indicated the role(s) that these authors had in your study. You can update author roles in the Author Contributions section of the online submission form.

The financial disclosure statement has been updated with the following:

“This work was supported by The Research Council of Norway [240207/F20]. The funders had no role in study design, data collection and analysis, decision to publish, or preparation of the manuscript. Elmira Flem is currently employed by Merck & Co., Inc., North Wales, PA, USA. The work on the current study was conducted by Dr. Flem under the previous affiliation. Elmira Flem’s current employer Merck & Co did not contribute financially or with financial support for salaries, as this work was conducted while Dr. Flem was employed at The Norwegian Institute of Public health.”

(2) Please also provide an updated Competing Interests Statement declaring this commercial affiliation along with any other relevant declarations relating to employment, consultancy, patents, products in development, or marketed products, etc. 

Within your Competing Interests Statement, please confirm that this commercial affiliation does not alter your adherence to all PLOS ONE policies on sharing data and materials by including the following statement: “This does not alter our adherence to PLOS ONE policies on sharing data and materials.” (as detailed online in our guide for authors http://journals.plos.org/plosone/s/competing-interests).

If this adherence statement is not accurate and there are restrictions on sharing of data and/or materials, please state these. Please note that we cannot proceed with consideration of your article until this information has been declared.

The competing interests declaration has been updated with: I have read the journal’s policy and the authors of this manuscript have the following competing interests: Elmira Flem is currently employed by Merck & Co., Inc., North Wales, PA, USA. The work on the current study was conducted by Dr. Flem under the previous affiliation at the Norwegian Institute of Public Health. This does not alter our adherence to PLOS ONE policies on sharing data and materials.

All other authors hereby declare that no other conflicts of interest exist. 

The data availability statement has been updated with: 

Because of ethical and legal restrictions to protect the identity and confidentiality of the participants, as the data contain sensitive information according to the Norwegian Personal Data Act, the underlying data are only available upon request from the Norwegian Institute of Public Health Institutional Data Access. Requests for access can be made to the data access committee represented by Anne-Marte Bakken Kran (Anne-MarteBakken.Kran@fhi.no)

4. One of the noted authors is a group or consortium: Norwegian Enhanced Pediatric Immunisation Surveillance (NorEPIS) Network

In addition to naming the author group, please list the individual authors and affiliations within this group in the acknowledgments section of your manuscript. Please also indicate clearly a lead author for this group along with a contact email address.

The individual authors and affiliations are now listed in the acknowledgements section, including an indication of lead author with contact email address.

 

Reviewers’ comments:

Reviewer’s Responses to Questions

Comments to the Author

1. Is the manuscript technically sound, and do the data support the conclusions?

Reviewer #1: Yes

Reviewer #2: Yes

2. Has the statistical analysis been performed appropriately and rigorously? 

Reviewer #1: Yes

Reviewer #2: I Don’t Know

3. Have the authors made all data underlying the findings in their manuscript fully available?

Reviewer #1: Yes

Reviewer #2: Yes

4. Is the manuscript presented in an intelligible fashion and written in standard English?

Reviewer #1: Yes

Reviewer #2: No

5. Review Comments to the Author

Reviewer #1: Dear colleagues I would like to thank you for the interesting research you have performed.

I was impressed by the high volume of data collected and the meticulous analysis performed.

Only trivial comments and questions:

1. page 11 first row of “Table 2: Participant characteristics and penalized maximum likelihood logistic regression of typed RSV cases. Excluding co-infections with one or more of Influenza, Metapneumovirus, Parainfluenza 1, 2, 3, 4 and Adenovirus, or cases with missing information on coinfections” - you have to correct “Inn-patients” with “inpatients”

Corrected

2. row 222-3 - “ Excluding 6 outliers with a LOS exceeding 12 days did not impact the result (not shown).” Please elaborate on these outliers with prolonged LOS - were they with nosocomial infections or preexisting severe conditions, etc?

Unfortunately we do not have any additional information about the length of stay for these 6 patients. The secondary analysis was performed to ensure that any discrepancies in the registration of admission or discharge date in these 6 patients did not influence the results. To make this more clear the following has been added to the manuscript:

“There were 6 patients with a LOS exceeding 12 days. To ensure that potential errors in the registration of the admission or discharge date did not impact the results a secondary analysis was conducted. Excluding these outliers did not impact the result (not shown).”

3. in page 17 row 241-244 “In the current study we investigated differences in disease severity, risk factors and comorbidities between RSV-A and RSV-B subtypes in young, hospitalized children in Norway during pre-pandemic period. Both RSV-A and B was found to co-circulate throughout all three study seasons, and no clear seasonal pattern was identified.”

Please elaborate on present research directions... Do you have any research ongoing on COVID-19 pandemic impact on RSV circulation and RSV subtypes? Because there is a plethora of published papers that are documenting major impact of SARS-CoV-2 phenomenon on circulation of certain viruses like influenza, RSV, etc and virtually no impact on Rhinovirus circulation. It would add value on Discussion section.

The following paragraph has been added with a corresponding change in the conclusion: 

“During the COVID-19 pandemic, many countries reported a disrupted seasonality of RSV, with little or no RSV circulating during the first years of the pandemic and with untypical out of seasonal peaks once disease control measures was lifted [1]. This disrupted and shifted seasonality of RSV was also observed in Norway during and after the pandemic [2]. Future studies are needed to elucidate if there are differences in the distribution of RSV subtypes after the pandemic.”

“[…] Future studies are needed to investigate how the distribution of RSV and RSV subtypes has changed during and after the COVIS-19 pandemic, and the potential impact this has for the prevention of RSV in the future.”

4. page 18 row 280 “ we only had one hospital contact pr. disease episode” - please rephrase this statement, or better the entire section because “contact pr. disease” is not included in your abbreviations section nor is a usual abbreviation.

The paragraph has been changed in accordance with the suggestion from the reviewer.

“Some patients are admitted several times during the course of an RSV infection. In order to avoid counting these patients multiple times we only used the first registered hospital contact during a disease episode”.

5. page 18 rows 285-288. Please elaborate on Norwegian national guidelines prophylaxis with palivizumab in premature children because the reference inserted [39] is not in English and can generate confusion. In your country there are implemented guidance for RSV prophylaxis in premature children of what gestational age and since when? Because study period is 2015-18 and reference provided seems to be from 2021.

The first General guidance in pediatrics in Norway was published in 2006, however subsequent updates have only been available in an electronic version that is continuously updated. The guidelines for the use of palivizumab were updated in 2014, 2021 and again in 2023. Although the 2014 and 2021 versions contained similar recommendations, there are some more substantial changes in the recent 2023 version (updated October 12. 2023). To avoid confusion regarding the current guidelines among Norwegian readers this paragraph has been changed to: 

“Restricting the analysis of premature children to extremely premature or very premature children according to the WHO definitions [41], could be warranted, however this was not possible in this study due to low numbers of premature children among the RSV-A and RSV-B cases.” 

The reference has been changed to the WHO fact sheet on preterm birth.

Reviewer #2: Abstract:

Authors reported the results of sex association with RSV-A or RSV-B, and the comorbidities prevalence among the two types. However, these results weren’t mentioned as an objective or an aim for this study. Results of age association with RSV-A or RSV-B was meant to be an objective of the study, but no result was reported in the results section.

The aim is described at the end of the introduction, mentioning both RSV-subtypes, disease severity, sex, age groups and comorbidities. Thank you for pointing out that this was not stated clearly enough. To make this clearer to the reader, this paragraph has been changed to:

“In the current study, we aimed to explore the associations between disease severity and RSV subtypes RSV-A and RSV-B in Norway between 2015 and 2018. We further aimed to describe the distribution of RSV subtypes pattern by season, sex, age groups and underlying comorbidities.”

At the end of the first paragraph of the result section it is stated “We found no significant differences between the proportion of RSV-A or B by age or sex, neither for the entire study population nor when restricting the analysis to inpatients only (Table 2 and 3).”

The results for age is also a part of both Table 2 and Table 3. 

Introduction: Introduction is well written. However, adding more details about the conducted study is important such as the duration and location of the study.

Country and time period is now stated as a part of the aim.

Methods:

- Exclusion criteria wasn’t clear, try to add a paragraph indicating the exclusion criteria of your study.

The following paragraph has been added to the methods section:

“Children >5 years of age or with residence outside of the hospital’s catchment area, children admitted more than 48 hours before enrollment, newborns that had not left the hospital and children referred from other hospitals or admitted for elective hospitalization, injury or social indication were excluded.”

- Did the studied patients were encountered at the emergency room or a clinical setting? Were they encountered in a tertiary hospital, secondary hospital, or primary care services?

The methods section has been changed to make this information more accessible. In addition the study setting has previously been described in detail in our previous publication and is referenced in this description:

" Children from 0-59 months of age presenting at hospital were prospectively enrolled through the Norwegian Enhanced Paediatric Immunisation Surveillance Network (NorEPIS). Both inpatients and outpatients were eligible. A detailed description including inclusion- and exclusion criteria, has been given previously [28]. Briefly, during three seasons from 2015 to 2018, RSV surveillance was implemented annually from week 40 to week 20 the following year in five major Norwegian primary hospitals, except for 2015 when surveillance was implemented from week 49.”

- Among the enrolled 2590 children, how many of them are below the age 12 months? And how many were between 12-59 months of age?

As one child could be enrolled several times e.g. once in season one and then again in season two or three, we do not give a specific age for each child, but we list the age during every hospital contact. This is shown in table 1. To make this clear to the reader the title of the table is changed to: “Age distribution of participants and distribution of RSV subtypes”.

The following has also been added to the methods section in the description of number of hospital contacts: “1449 children were <12 month old and 1276 children were 12-59 months old”

- Results:

- Results (Table 2, 3): Tables are confusing, and legends does not differentiate them. Also, try to use some in text explanation of these two tables. Furthermore, what type of respiratory support? What are the comorbidities?

Table 2 shows the analysis and distribution of all participants tested for RSV-A and RSV-B, while Table 3 restricts the analysis to inpatients only. To make this distinction clearer the title of table 2 has been changed to “Participant (inpatients and outpatients combined) characteristics and penalized maximum likelihood logistic regression of typed RSV-cases. Excluding co-infections with one or more of Influenza, Metapneumovirus, Parainfluenza 1, 2, 3, 4 and Adenovirus, or cases with missing information about coinfections”

The comorbidities are listed as a footnote at the bottom of the tables: “† Including trisomy 21, neuromuscular, impairment, congenital heart disease, pulmonary disease, BPD, immunodeficiency, and cancer.” 

Respiratory support is defined in the methods section, under Risk groups and severity. To make this information more accessible we have also added the definition as footnote in Table 1 and Table 2

- In the severity section add the p-value for significance.

The p-value is now stated in the severity section.

- More in-text elaboration of the results is needed. However, the first part of the result and the tables are confusing and weren’t clearly delivered.

I am sorry. I do hope it has improved with the changes we have added.

Discussion

- A summarization of the main results with its significant level wasn’t included.

The summarization of main results at the start of the discussion has been elaborated. 

- A comparison of the virus seasonality with the previous findings of the virus pattern in Norway needs to be done.

To our knowledge this is the first study to compare and describe the seasonality of RSV-A and RSV-B in Norway. The following paragraph has been added to the discussion:

“The seasonal patterns of the RSV subtypes in circulation in Norway, has to our knowledge, not previously been described. However, the overall total RSV seasonality described is similar to previous reports from Norway [34] and is presented in detail in our previous publication [28].”

- The study results of Risk factors and comorbidities weren’t compared with other studies.

To our knowledge, this is the first study to investigate the possible association between different comorbidities and RSV subtypes. To emphasize this the following paragraph has been added.

“To our knowledge, this is the first study to investigate if there is an association between the presence of comorbidities and RSV subtypes. However, one study found that a stronger association between RSV-B infection and a family history of asthma compared to RSV-A. However, this association also varied between RSV-A genotypes, with some genotypes showing a similar association to asthma as RSV-B [42]”

- Overall and apart from the severity section, the discussion is unorganized, poorly written, and lacks the inclusion of supporting evidence and other studies.

I am sorry. I do hope it has improved with the changes and additional references we have added.

1. Stein, R.T. and H.J. Zar, RSV through the COVID-19 pandemic: Burden, shifting epidemiology, and implications for the future. Pediatric Pulmonology, 2023. 58(6): p. 1631-1639.

2. Norwegian Institute of Public Health, Epidemier av luftveisinfeksjoner i Norge vinteren 2023-24: risiko, scenarioer og håndtering. Available from: https://www.fhi.no/publ/statusrapporter/risikovurdering-for-luftveisinfeksjoner/, 2023.

---

## [Editor Report · Decision Letter 1]

20 Jan 2024

No association between disease severity and Respiratory syncytial virus subtypes RSV-A and RSV-B in hospitalized young children in Norway

PONE-D-23-21221R1

Dear Dr. Bøås,

We’re pleased to inform you that your manuscript has been judged scientifically suitable for publication and will be formally accepted for publication once it meets all outstanding technical requirements.

Kind regards,

Victor Daniel Miron

Academic Editor

PLOS ONE

---

## [Editor Report · Acceptance letter]

24 Feb 2024

PONE-D-23-21221R1 

PLOS ONE

Dear Dr. Bøås, 

I'm pleased to inform you that your manuscript has been deemed suitable for publication in PLOS ONE. Congratulations! Your manuscript is now being handed over to our production team.

Kind regards, 

on behalf of

Dr. Victor Daniel Miron 

Academic Editor

PLOS ONE